# Unraveling Complexity: Singular Value Decomposition in Complex Experimental Data Analysis

**Judith F. Stein, Aviad Frydman and Richard Berkovits⋆**

Department of Physics, Jack and Pearl Resnick Institute, Bar-Ilan University,
Ramat-Gan 52900, Israel

⋆ richard.berkovits@biu.ac.il

## Abstract

Analyzing complex experimental data with multiple parameters is challenging. We propose using Singular Value Decomposition (SVD) as an effective solution. This method, demonstrated through real experimental data analysis, surpasses conventional approaches in understanding complex physics data. Singular value amplitudes and vectors distinguish and highlight various physical mechanisms and scales, revealing previously challenging elements. SVD emerges as a powerful tool for navigating complex experimental landscapes, showing promise for diverse experimental measurements.

## 1 Introduction

Singular value decomposition (SVD) finds extensive applications, primarily in data compression [1–4] and machine learning [5, 6]. While physicists recognize its crucial role in

defining entanglement entropy [7], its utilization in analyzing and interpreting experimental data has often been confined to niche applications [8–11]. However, SVD holds significant potential for the analysis of complex experimental data, particularly data arising from distinct physical mechanisms concurrently influencing the experimental results. By adjusting a control parameter, one can modulate these mechanisms to varying degrees. Leveraging SVD eliminates the need for prior assumptions in modeling the contributions of these mechanisms to the measurements.

In recent numerical studies, researchers have employed SVD analysis to examine the numerically calculated energy spectra of complex chaotic quantum systems [12–22]. The energy spectra of quantum chaotic systems are influenced by both universal and system-specific features, presenting a challenging task commonly referred to as "unfolding" within the field. Various unfolding methods have been utilized, and SVD has demonstrated a distinct advantage in revealing universal properties of the spectrum, particularly on larger energy scales.

SVD, a linear algebra technique, allows the rewriting of any matrix with dimensions $M \times P$ as a sum of amplitudes (termed singular values) multiplied by an outer product of two vectors, where the number of terms is determined by $\min(M, P)$. Details of this process will be discussed in Sec. 2. The singular values, being positive, can be ordered by size, enabling the approximation of the original matrix through a sum over a reduced number of the larger terms, significantly fewer than $\min(M, P)$.

Why does this mathematical exercise matter for experimental measurements? After all, most experimental data isn't structured like a matrix. However, if the results of the measurements depend on two parameters where at least one of them is equidistantly sampled (or interpolated), one can organize the data by performing $M$ measurements of one parameter where for each such measurement the second parameter is measured $P$ times (see Fig. 1a), into an $M \times P$ matrix.

We will showcase the effectiveness of the SVD model through experimental measurements of differential current conducted on both one- and two-dimensional arrays of superconducting dots on a graphene substrate. By sweeping the dc voltage at various gate voltages, the measured conductivity exhibits a pronounced dependence on both bias and gate voltages. Oscillations in relation to the dc voltage, with seemingly distinct periods in different regions, are observed. Through SVD analysis, we aim to untangle this intricate data, gaining valuable insights into the dependence of experimental measurements on the two parameters.

The paper unfolds in the subsequent sections. In Sec. 2, we delve into an exposition of the SVD method, elucidating its application to data analysis. Sec. 3 is dedicated to detailing the experiment and the acquired experimental data, along with speculative insights into the underlying physics. Motivated by the discernible oscillations in the data concerning the dc voltage, we embark on Fourier analysis in an attempt to glean an interpretation; however, the results prove inconclusive. Subsequently, in Sec. 4, we harness the power of SVD analysis, revealing its capacity to yield a markedly clearer interpretation of the data. The final section (Sec. 5) undertakes a discussion on the broader application of SVD analysis to other experimental measurements.

## 2   The SVD method

As discussed in the introduction, the initial step in applying SVD analysis involves transforming the experimental measurement $X(U, V)$, dependent on parameters $V$ and $U$, into a matrix. Without loss of generality, let us assume that $V$ is swept (or interpolated)

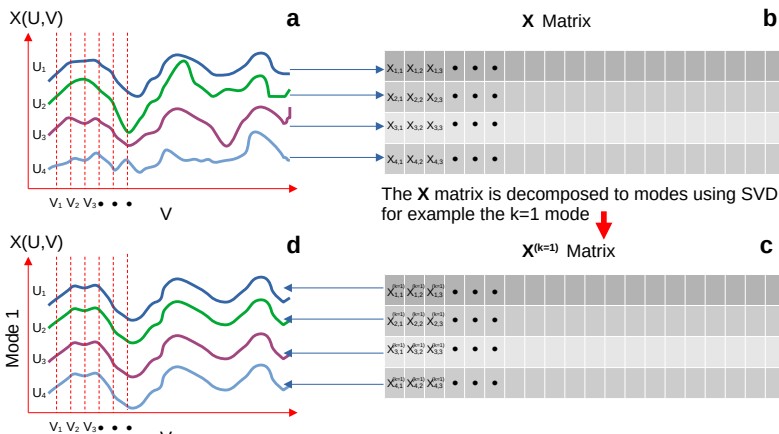

Figure 1: **The SVD procedure.** A schematic cartoon of the SVD procedure. In (a), a physical observable $X$, dependent on two parameters $U$ and $V$, is measured. The procedure involves setting $U_i$ ($i = 1, 2, \dots$) while changing $V$, resulting in the curves for $X(U_i, V)$ illustrated in the graph. In (b), to represent the data as a matrix $\mathbf{X}$, $V$ is discretized into $V_j$, and each value of $X(U_i, V_j)$ is inserted as the matrix element $X_{i,j}$. Thus, each row corresponds to the measurements for a given value of $U_i$. The SVD procedure is applied, yielding a series of matrices $\mathbf{X}^{(k)}$, with the original matrix expressed as a sum of modes $\mathbf{X} = \sum_k \sigma_k \mathbf{X}^{(k)}$, where $\sigma_k$ is the singular value amplitude, and the modes are ordered by magnitude from the largest. In (c), the matrix for the largest mode, $k = 1$, is represented. Due to the structure of the SVD procedure (see text), each matrix element in $\mathbf{X}^{(k=1)}$ is equal to $\vec{U}_i^{(k=1)} \vec{V}_j^{(k=1)}$. Thus, each row is equivalent to the same vector $\vec{V}^{(k=1)}$ multiplied by a different constant $\vec{U}_i^{(k=1)}$. This relationship is illustrated in the plot (d), corresponding to the curves $X(U_i, V)$ for the first mode.

at equidistant increments, such that $V_j = j\Delta V$ for $j = 1, 2, \ldots P$. On the other hand, the second parameter, $U$, may not necessarily increase at equidistant intervals or even be ordered. It suffices for $U$ to be set at $M$ different values, denoted as $U_i$. Consequently, a $M \times P$ matrix $\mathbf{X}_{ij} = X(U_i, V_j)$ can be constructed as schematically illustrated in Fig. 1.

In the SVD procedure, the matrix $\mathbf{X}$ is expanded as a sum of amplitudes $\sigma_k$ multiplied by $M \times P$ matrices $\mathbf{X}^{(k)}$. These matrices are constructed by an outer product of two vectors $\vec{U}_i^{(k)}$ and $\vec{V}_j^{(k)}$ of sizes $M$ and $P$, respectively. Explicitly, $\mathbf{X}$ is decomposed into $\mathbf{X} = \mathbf{U}\mathbf{\Sigma}\mathbf{V}^T$, where $\mathbf{U}$ and $\mathbf{V}$ are $M \times M$ and $P \times P$ matrices, respectively, and $\mathbf{\Sigma}$ is a diagonal matrix of size $M \times P$ with a rank $r = \min(M, P)$. The $r$ diagonal elements of $\mathbf{\Sigma}$ are the singular values (SV) amplitudes $\sigma_k$ of $\mathbf{X}$. These SVs are positive and can be ordered by magnitude as $\sigma_1 \geq \sigma_2 \geq \ldots \geq \sigma_r$. As discussed, $\mathbf{X}$ can be expressed as a series of matrices $\mathbf{X}^{(k)}$, i.e., $\mathbf{X}_{ij} = \sum_{k=1}^{r} \sigma_k \mathbf{X}_{ij}^{(k)}$, where $\mathbf{X}_{ij}^{(k)} = \mathbf{U}_{ik}\mathbf{V}_{jk}^T = \vec{U}_i^{(k)}\vec{V}_j^{(k)}$. The sum of the first $m$ modes provides an approximation $\tilde{\mathbf{X}} = \sum_{k=1}^{m} \sigma_k \mathbf{X}^{(k)}$ to $X$, representing the minimal departure between the approximate measurements, $\tilde{\mathbf{X}}$, obtained using $m(M + P + 1)$ independent variables compared to the full energy spectrum, which requires $MP$ variables. This forms the basis for the use of SVD as a data compression method. Since, for most cases (including those discussed here), the SVs drop rapidly as a function of $k$, a good approximation of $\mathbf{X}$ is achieved. Indeed, examining the SVs as a function of $k$, typically involving a Scree plot plotting $\lambda_k = \sigma_k^2$ vs. $k$ on a logarithmic scale, serves as the first step in analyzing the data.

The SV amplitudes, $\sigma_k$, corresponding to significant modes (typically with $k \sim O(1)$), along with the associated vectors $\vec{U}^{(k)}$ and $\vec{V}^{(k)}$ for these modes, play a crucial role in interpreting experimental data. This importance can be illustrated through an analogy with one of the most widely used experimental data analysis methods, the Fourier transform. In the case of a Fourier transform, the experimental results $X(U_i, V_j)$ can be expressed as $\sum_{k_i, k_j} f_{k_i, k_j} \sin(k_i)\sin(k_j)$. Superficially, the structure bears similarity to the SVD sum, as both involve an amplitude multiplied by two vectors or functions. In both methods, the goal is to identify amplitudes significantly larger than others to characterize the data. Furthermore, the general dependence of these amplitudes on the mode or frequency can offer insights into the overall characteristics of the system, such as the presence of $1/f$ noise.

Nonetheless, significant distinctions exist. The SVD sum involves just $r = \min(M, P)$ amplitudes, a stark contrast to the $MP$ amplitudes present in the Fourier transform. This reduction in the number of terms in the SVD sum arises because, unlike the fixed vectors involved in the outer multiplication of the Fourier transform, the vectors in SVD are optimized to achieve the best fit with a minimal number of modes. Consequently, in contrast to the Fourier transform, valuable insights are gained not only from the amplitudes but also from the optimized vectors $\vec{U}^{(k)}$ and $\vec{V}^{(k)}$ associated with contributing modes.

In the subsequent sections, we will elaborate on these somewhat vague ideas by implementing them using concrete experimental data. This data is derived from conductance measurements performed on one- and two-dimensional superconducting grain arrays deposited on graphene.

## 3 Experimental results

We analyze results obtained on single-layer-graphene (SLG) films decorated by ordered arrays of disordered superconducting indium oxide ($InO$) dots. We compare two geome-

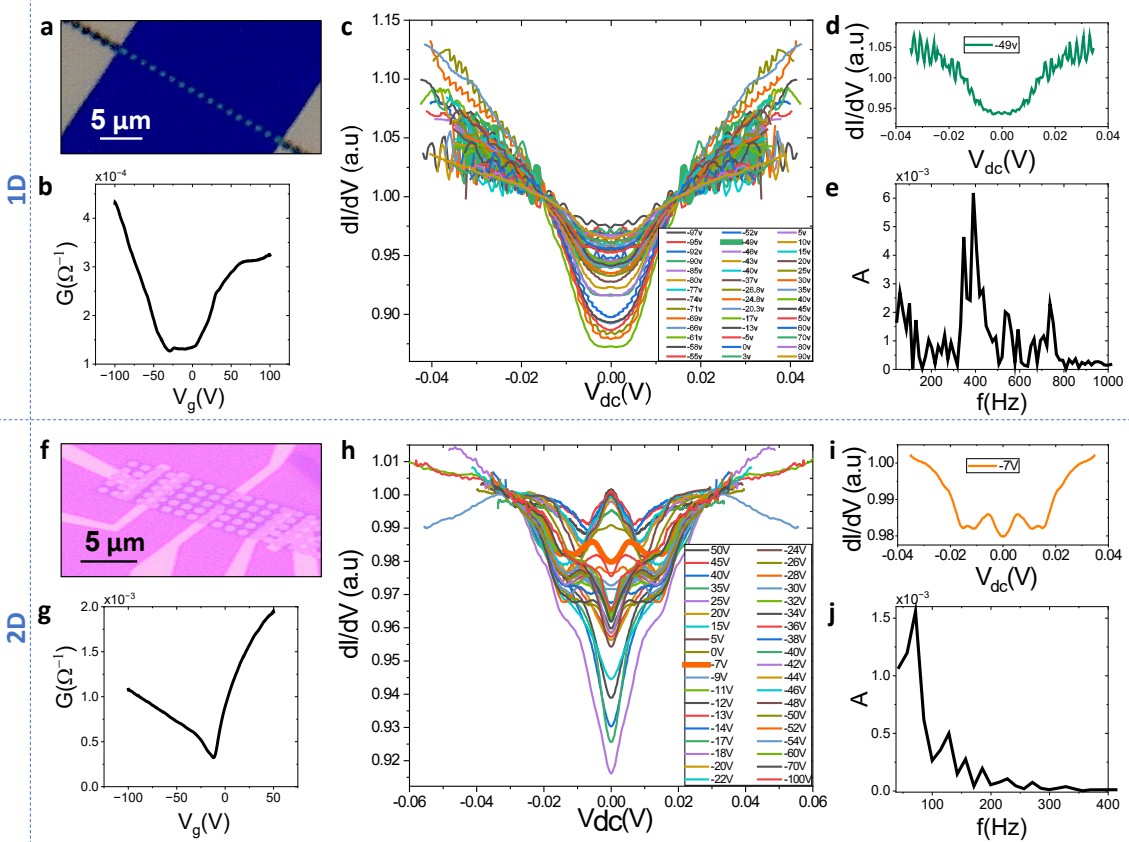

Figure 2: **Raw data for the 1D (top panels) and 2D (bottom panels) samples.** **(a)** and **(f)** show optical microscope images of a 1D and 2D SLG/SC-dot-array configurations respectively. The respective conductance, $G$, versus gate voltage, $V_g$, curves are depicted in **(b)** and **(g)** showing a conductance dips at the Dirac points of the underlying graphene. Corresponding sets of differential conductance, $dI/dV$, versus bias voltage, $V_{dc}$ measurements at different gate voltages, are shown in **(c)** and **(h)**. Typical $dI/dV - V_{dc}$ curves are singled out in **(d)** and **(i)** for which FT analysis are shown in **(e)** and **(j)**.

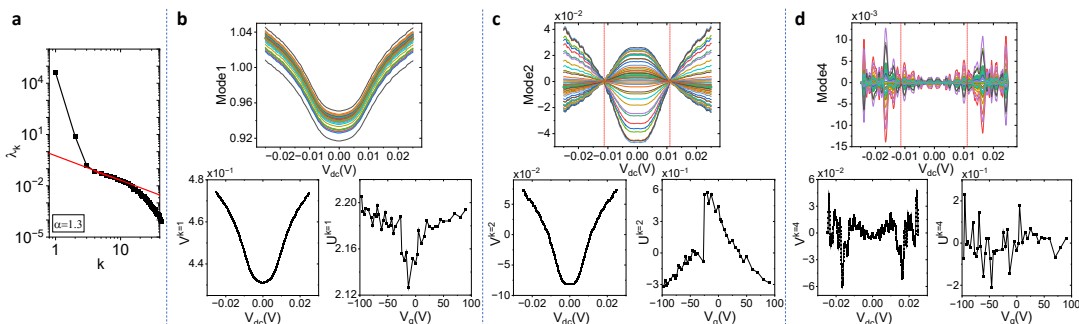

Figure 3: **SVD analysis of the 1D sample. (a)** A scree plot of SV amplitude squared ($\lambda_k = \sigma_k^2$) as function of the mode number $k$ for the 1D sample. The first mode is orders of magnitude larger than the rest, while the second mode deviates from the power-law behavior seen for larger modes for which $\lambda_k \sim k^{-1.3}$. **(b,c,d)** Top panels: the contribution of the first mode ($k = 1$), second mode ($k = 2$) and fourth mode ($k = 4$) respectively to the measured data. Note that a distinct feature of the second mode, seen for both 1D and 2D samples is the fact that they intersect at a distinct value of voltage $V'_{dc} = \pm 12mV$ for the 1D sample, and $V_{dc} = \pm 9mV$ for the 2D sample, indicated by the dashed red lines. For $k = 4$ the values connected to the superconducting $V'_{dc}$ are depicted by the dashed red line. Bottom panels: the vector $\vec{V}^{(k=1/2/4)}$ (left) and $\vec{U}^{(k=1/2/4)}$ (right). The curves in the main panels are calculated by multiplying the SV amplitude times $\vec{V}^{(k=1/2/4)}$ by the appropriate $\vec{U}^{(k=1/2/4)}_{V_g}$ for each curve.

107 tries: (i) A one-dimensional row of 17 sequential dots shown in Fig. 2a (1D sample) and
108 (ii) a two-dimensional array of $16 \times 5$ dots shown in Fig. 2f (2D sample). The SLGs
109 were fabricated either by flake-exfoliation or CVD growth on top of a $Si/SiO$ substrate.
110 The graphene layers were etched to create rectangles with dimensions of $1\mu m \times 18\mu m$
111 (1D) and $17\mu m \times 6\mu m$ (2D) using standard lithography followed by $RIE$ process. Suit-
112 able $Cr/Au$ contacts were deposited on the samples for electric measurements and an
113 additional electrode was fabricated on the back side of the $Si$ substrate to act as a gat-
114 ing electrode. The superconducting dot arrays were prepared by e-beam evaporation of
115 $50nm$ thick $InO$ film patterned to produce $1\mu m$ diameter dots with $200nm$ inter-dot
116 distance. The $InO$ was e-beam evaporated at a partial oxygen pressure of $\approx 1 \times 10^{-5}$
117 mbar, resulting in disordered superconducting film with a $T_c$ of $\sim 3.5K$. All electronic
118 measurements were conducted in a $He_3$ system at $T = 0.33K$.

119     Fig. 2 c,h show differential conductance versus bias voltage $(dI/dV - V_{dc})$ curves
120 at different gate voltage, $V_g$, for a 1D and a 2D sample. It is evident that the data for
121 both the 1D and 2D samples is rather complex. The measurements reveal an intricate
122 dependence on both $V_{dc}$ and $V_g$. As illustrated in Fig. 2 b,g, which show the conductance,
123 $\lim_{V_{dc}\to 0} G = dI/dV$ plotted as a function of $V_g$, it is observed that $G$ exhibits a dip in
124 the proximity of $V_g \sim 0$. Conversely, at higher values of $V_{dc}$, $V_g$ has a weaker influence
125 on $dI/dV$. Oscillations are observed at certain values of $V_g$, whereas at others, they are
126 less pronounced. Furthermore, these oscillations appear to depend on $V_{dc}$.

127     The complex structure of the curves is expected to be a result of three main contribu-
128 tions:

129 1. A depletion of the electronic DOS around the Fermi level due to the Altsuler-Aronov
130 (AA) mechanism of electron-electron interactions in disordered films [23].
131 2. A Superconducting gap, $\Delta$ in the graphene regions below the $InO$ dots due to the
132 proximity effect [24], each with an expected bias scale of $\Delta_{InO} \approx 0.7mV$ [25].

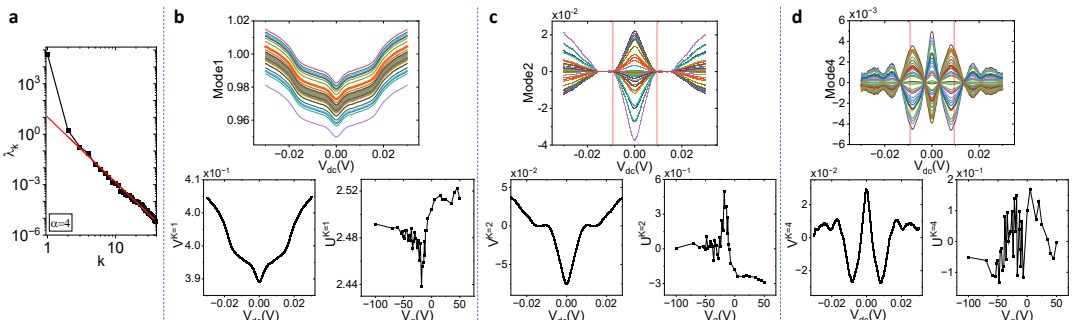

Figure 4: **SVD analysis of the 2D sample.** similar to those presented in Fig. 3. Note that for this sample, $\lambda_k \sim k^{-4}$.

3. Electronic quantum interference effects resulting from the periodic structure of superconducting-normal region interfaces. These effects depend on the Fermi velocity of graphene, $v_F \approx 10^6$ m/s, and the inter-dot distance, $\approx 200 nm$, leading to an expected period as function of $V_{dc}$ of $\approx 2mV$.

In order to appropriately analyze these results, one would like to decompose the different physical contributions to the data. A naive way to do so would be to employ a simple Fourier transform. However, the intricacies involved largely rule out a 2D Fourier transform of both $V_{dc}$ and $V_g$. Even when attempting a Fourier transform solely for $V_{dc}$ at a fixed $V_g$ where oscillatory behavior is unmistakable, no distinct peak in frequency is evident (see Fig. 2 e,j). This lack of clarity in frequency peaks makes it challenging to draw meaningful conclusions from the Fourier transform analysis. In addition, such analysis method requires separate calculation for each individual $V_g$ value in an attempt to identify repeating patterns. Clearly, a more useful and efficient analysis tool is required.

# 4    SVD Analysis

Hence, we apply SVD analysis to the experimental data presented in the previous section (Sec. 3). As outlined in Sec. 2, the initial step involves examining the behavior of the SV amplitudes. In Figs. 3a and 4a, a scree plot illustrates the squared SV amplitudes $(\lambda_k = \sigma_k^2)$ in relation to the mode number $k$. Notably, the largest SV amplitude $(k = 1)$ is orders of magnitude greater than subsequent modes for both samples. Beyond $k = 3$, a power-law behavior emerges. Specifically, the 1D chain exhibits a power law described by $\lambda_k \sim k^{-1.3}$ (Fig. 3a), while the 2D sample follows a steeper power law, $\lambda_k \sim k^{-4}$ (Fig. 4a). This disparity in power laws is significant; as demonstrated in the appendix of Ref. [15], a power law of $\lambda_k \sim k^{-1}$ corresponds to $1/f$ noise. Consequently, modes $k = 3-15$ for the 1D sample appear to align with characteristics of $1/f$ noise. In contrast, the 2D sample seems well-characterized by the initial few modes, as the contribution from subsequent modes rapidly diminishes. This observation is reinforced by noting that measurements of the 1D sample exhibit greater noise compared to those of the 2D sample (Fig. 2).

Now, let us delve into an examination of the contributions from individual modes. The contributions of the first mode $(k = 1)$ to the measured data are shown in Fig. 3b for the 1D sample and Fig. 4b, for the 2D sample, along with the associated vectors $\vec{V}^{(k=1)}$ and $\vec{U}^{(k=1)}$. The differential conductance, $dI/dV$, as a function of $V_{dc}$ for various values of $V_g$ is plotted, where the various values are coded with the same color code as in Fig. 2. As discussed in Sec. 2, $\mathbf{X}^{(k=1)} = \sigma_1 \vec{V}^{(k=1)} \otimes \vec{U}^{(k=1)}$. The outer

multiplication between these two vectors has a transparent interpretation. Specifically, the vector $\vec{V}^{(k=1)}$ captures the first mode's dependence of the differential conductance, $dI/dV$, on $V_{dc}$. Consequently, the vector $\vec{V}^{(k=1)}$ is multiplied by the term of the vector $\vec{U}^{(k=1)}$ that corresponds to the appropriate value of $V_g$. This relationship is visually evident in the main panels of Figs. 3b and 4b, where the multiplication of $\vec{V}^{(k=1)}$ by the corresponding value of $\vec{U}^{(k=1)}$ is plotted for each term of $\vec{U}^{(k=1)}$, i.e., for each value of the gate voltage $V_g$.

Hence, the first mode derived from the SVD provides an overall insight into the behavior of the differential conductance. For our samples, we associate this gross feature with AA depletion in disordered metals. AA depletion manifests in a logarithmic increase in the differential conductance, which is truncated at low voltage due to temperature. Indeed, in the case of the 1D sample, the first mode vector $\vec{V}^{(k=1)}$ exhibits a broad minimum around $V_{dc} = 0$, followed by a logarithmic increase. For the 2D sample, the behavior is more intricate, and a sharp minimum at $V_{dc} = 0$ appears, revealing a more distinct structure that needs further explanation. It's noteworthy that, unlike modes in the Fourier transform, SVD tailors its vectors to the specific measurements, as exemplified by the contrast between $\vec{V}^{(k=1)}$ for the 1D and 2D samples.

Additionally, while $\vec{V}^{(k=1)}$ captures the fundamental features of the experiment for the 1D sample, it misses notable features observed in the 2D sample, such as the transformation of the minimum at $V_{dc} = 0$ into a maximum for certain values of $V_g$. An examination of the behavior of $\vec{U}^{(k=1)}$ as a function of $V_g$ reveals a close correlation with the behavior of $G$, as depicted in Fig. 2 b,g.

Next we turn to the second mode of the SVD analysis. The mode is plotted in Figs. 3c and Fig. 4c. A very clear feature of $\mathbf{X}^{(k=2)}$ of both samples is that distinct regions of behavior as function of $V_{dc}$ are revealed. All curves cross at two values of $V'_{dc} = \pm 12mV$ for the 1D sample and at $V'_{dc} = \pm 9mV$ for the 2D sample. These values of $V'_{dc}$ correspond to the estimation of the superconducting gap in these systems, and they are unequivocally revealed by the second mode of the SVD. Considering the simpler 1D, which includes 17 junctions (dots) in series, one can expect to observe structure at $\Delta_{InO} \times 17 = 11.9mV$. Remarkably, this aligns exactly with the point where the curves of the second mode of the 1D sample intersect. For the 2D sample the shortest path across the sample is of 12 junctions, corresponding to $\Delta_{InO} \times 12 = 8.4mV$, not far from the estimation garnered from the width of the second mode.

The higher modes expose more intricate effects on the differential conductance, evident in the oscillations with respect to $V_{dc}$. Complicating the analysis is the observation that these oscillations seem to exhibit a different period within the region of the superconducting gap compared to outside of it. Moreover, this phenomenon is more pronounced for specific values of $V_g$. As illustrated in Figs. 3d and Fig. 4d, where one of the typical higher modes ($k = 4$) is presented, it is apparent that the amplitude and frequency of the oscillations differ for $|V_{dc}| < V'_{dc}$ compared to $|V_{dc}| > V'_{dc}$. In the 1D sample case, those frequencies found to be $2.5mV$ for $|V_{dc}| < V'dc$, inside the superconducting gap, and $1.9mV$ for $|V_{dc}| > V'_{dc}$, outside of it. Other high modes, such as $k = 3, 5, 6$, show a similar, although somewhat noisier periodicity. As noted above, such a voltage scale is expected for electronic interference effects due to the dot periodicity. For the 2D, which includes more than one single dot periodicity, the electronic interference effects are washed out and the oscillations are much slower, of order of $10mV$ which fits the gap energy.

## 5   Conclusion

In this work, we demonstrated the strength of the SVD technique, beyond its conventional applications, to assist in analyzing complex physics experimental data. We showed that the SV amplitudes and the different modes effectively separate and highlight distinct physical mechanisms that construct the results, which were otherwise difficult to isolate . Hence, the SVD is found to be an excellent tool for navigating through experimental data complexities, successfully reducing the dimensionality while preserving crucial information. It stands as a valuable asset for sophisticated experimental data analyses and holds further promise for unveiling valuable insights of real physics properties.

The potential of utilizing the SVD method for experimental data is vast, as it can essentially be employed to any experiment where data depends on two variables. For instance, it may be a most useful tool for analyzing mesoscopic systems where resistivity as a function of voltage and magnetic fields exhibits repeatable fluctuations with no clear period [26]. Similarly, optical spectra often shows non trivial structure as a function of e.g., wavelength and temperature. Alternatively, SVD may be effective for analyzing scanning images of a physical property as a function of lateral X and Y axes where one would like to deconvolute real physics from scanning noise and effects of the scanning probe kernel. SVD has also recently been used in network data analysis [27]. These are few examples for the immense potential of SVD applications in experimental physics data analysis. Its utility extends far and wide, making SVD an invaluable asset for diverse scientific disciplines.

**Author contributions**     J.F.S and A.F. carried out the experiments. R.B. carried out the theoretical analysis. All the authors discussed the results and jointly wrote the manuscript.

**Funding information**     AF and JS would like to acknowledge the support by the Ministry of Science and Technology, Israel and by the Israel Science foundation ISF grant number 1499/21.

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
