# Peer review of "Unraveling Complexity: Singular Value Decomposition in Complex Experimental Data Analysis"

_SciPost Physics Core_

## Round 1 · Referee Report · Anonymous (Referee 1) · 2024-6-13

Report

The authors offer a fresh look at the one-century-old mathematical method: Singular Value Decomposition (SVD). In contrast to the typical presentation for data analysis, where SVD is performed on a data matrix representing n samples (observations or measurements) of p variables (features), the authors consider the data matrix as parametrized by two parameters corresponding to rows and columns.

After highlighting the extensive applications of Singular Value Decomposition (SVD) in data compression and machine learning, while noting its underutilized potential in analyzing complex experimental data influenced by distinct physical mechanisms, the authors demonstrate the strength of SVD by effectively analyzing complex physics experimental data. The authors analyze single-layer-graphene films with arrays of superconducting indium oxide dots in one-dimensional (17 dots) and two-dimensional (16×5 dots) geometries. Fabricated via flake-exfoliation or CVD growth, the films were measured at 0.33K. Differential conductance versus bias voltage showed complex dependencies on both bias and gate voltages, influenced by electron-electron interactions, a superconducting gap, and quantum interference effects. Fourier transform analysis proved inadequate, necessitating a more effective analysis tool.

SVD isolated distinct physical mechanisms within the results, which are otherwise difficult to separate, while reducing dimensionality and preserving crucial information. The authors' application of SVD to real experimental measurements highlighted its ability to reveal valuable insights into the underlying physics, providing a clearer understanding of the data. These results underscore SVD's effectiveness and versatility, establishing it as a powerful tool for complex experimental data analysis.

The paper makes a valuable contribution by demonstrating the versatility and effectiveness of SVD in analyzing experimental data. The authors' approach and thorough analysis provide valuable insights into the underlying physics. The paper should be accepted after minor corrections as mentioned below.

Requested changes

  1. Recent advances in data analysis using SVD should be mentioned, especially the seminal work of Gavish and Donoho on the recovery of low-rank matrices from noisy data:
  2. Gavish, M., & Donoho, D. L. (2014). The optimal hard threshold for singular values is $4/\sqrt {3} $. IEEE Transactions on Information Theory, 60(8), 5040-5053.
  3. Gavish, M., & Donoho, D. L. (2017). Optimal shrinkage of singular values. IEEE Transactions on Information Theory, 63(4), 2137-2152.

  4. One important missing contextual remark is the use of SVD in statistics. Performing PCA on a dataset is equivalent to performing SVD on the centered data matrix, with the principal components and directions derived directly from the SVD's components. However, it should be explained that the point of view adopted in the paper differs significantly from the traditional interpretation of rows and columns in data matrix.

  5. The rationale for assuming that 'V is swept (or interpolated) at equidistant increments' is not explicitly stated and is not mentioned when analyzing the experimental data. The authors should clarify this assumption and indicate what happens when it is not satisfied.

  6. The term 'singular values amplitudes' seems atypical. In most mathematical and data analysis literature, 'singular values' is the preferred and widely accepted term.

  7. Figures:

  8. Figure 1: The plot and text in Figure 1 are too small compared to the text, which complicates its comprehension.
  9. Figure 2 (e,f): This is not "FT analysis" but rather the "power spectrum (or power spectral density) of the data obtained by FT analysis."

  10. There is inconsistency in the shape of vectors. The authors should mention when the vectors are rows and when they are columns, which would avoid any confusion when writing ( U_iV_j ). Only the lengths of ( U_i ) and ( V_j ) are mentioned while clearly they are interpreted as matrices in the paper. Note that the "outer product," mentioned on line 67 and displayed on line 166, is most often taken between two column vectors.

  11. Line 73: It might be better to mention that this is a series of rank-1 matrices, better stressing that the matrices ( X^{(k)} ) are basic elements analogue to the Fourier modes.

  12. Line 97: The optimality is a classical result in linear algebra, often called the "Eckart-Young theorem."

Recommendation

Ask for minor revision

  • validity: high
  • significance: good
  • originality: good
  • clarity: high
  • formatting: good
  • grammar: good

Author:  Richard Berkovits  on 2024-06-16  [id 4570]

(in reply to Report 1 on 2024-06-13)

We thank the referee for their careful reading of the manuscript and accept all their requested changes.

---

## Round 1 · Referee Report · Anonymous (Referee 2) · 2024-7-22

Strengths

1) The introduction and motivation is excellent and well written 2) I am actually interested in the application of the SVD for this purpose 3) The authors guide this to some interesting potential use cases 4) I think this paper could be highly cited and a good starting point for some groups looking for alternative analysis techniques in a wide variety of areas

Weaknesses

1) The paper has a hard drop into the rigour of the method which I think will prevent some readers from syntehsizing it 2) There are few general guidelines beyond the use case presented here for other groups

Report

I think the paper in general can be published. It suffers from trying to be a very general idea but focusing mostly on a specific implementation without having many example or guide steps for how to use the SVD in new cases. The specific case used has the advantage of having a decreasing singular value decomposition, which should be expected for physically relevant cases, but the broad class of where this algorithm can be applied will need to come from other sources.

The paper is extremely thorough for what is covered.

Requested changes

I think the authors did a good job responding to the previous referee and I have no major outstanding comments on it.

Recommendation

Publish (meets expectations and criteria for this Journal)

---

## Editorial Decision

resubmitted